environmental science

network growth, sustainable transport, bicycle infrastructure

**Authors for correspondence:**
Luis Guillermo Natera Orozco
e-mail: natera_luis@phd.ceu.edu
Michael Szell
e-mail: misz@itu.dk

# Data-driven strategies for optimal bicycle network growth

Luis Guillermo Natera Orozco[1], Federico Battiston[1], Gerardo Iñiguez[1,2,3] and Michael Szell[4,5,6]

[1]Department of Network and Data Science, Central European University, 1100 Vienna, Austria
[2]Department of Computer Science, Aalto University School of Science, 00076 Aalto, Finland
[3]Centro de Ciencias de la Complejidad, Universidad Nacional Autonóma de México, 04510 CDMX, Mexico
[4]NEtwoRks, Data, and Society (NERDS), IT University of Copenhagen, 2300 Copenhagen, Denmark
[5]ISI Foundation, 10126 Turin, Italy
[6]Complexity Science Hub Vienna, 1080 Vienna, Austria

 LGNO, 0000-0001-6574-0614; FB, 0000-0001-9646-6232;
GI, 0000-0001-7181-5520; MS, 0000-0003-3022-2483

Urban transportation networks, from pavements and bicycle paths to streets and railways, provide the backbone for movement and socioeconomic life in cities. To make urban transport sustainable, cities are increasingly investing to develop their bicycle networks. However, it is yet unclear how to extend them comprehensively and effectively given a limited budget. Here we investigate the structure of bicycle networks in cities around the world, and find that they consist of hundreds of disconnected patches, even in cycling-friendly cities like Copenhagen. To connect these patches, we develop and apply data-driven, algorithmic network growth strategies, showing that small but focused investments allow to significantly increase the connectedness and directness of urban bicycle networks. We introduce two greedy algorithms to add the most critical missing links in the bicycle network focusing on connectedness, and show that they outmatch both a random approach and a baseline minimum investment strategy. Our computational approach outlines novel pathways from car-centric towards sustainable cities by taking advantage of urban data available on a city-wide scale. It is a first step towards a quantitative consolidation of bicycle infrastructure development that can become valuable for urban planners and stakeholders.

## 1. Introduction

Most modern cities have followed a car-centric development in the twentieth century [1] and are today allocating a privileged

**Figure 1.** (Map plots, left) Networks representing various layers of transport infrastructure (pedestrian paths, bicycle paths, rail lines and streets) for Copenhagen and London, with data from OpenStreetMap. (Right) Connected component size distribution $P(N_{cc})$ as a function of the ranking of the component for all considered network layers and cities. All layers are well connected except the bicycle layer: Copenhagen has 321 bicycle network components despite being known as a bicycle-friendly city, while London's bicycle layer is much more fragmented, featuring over 3000 disconnected components. Copenhagen's largest connected bicycle component (leftmost data point) spans 50% of the network, but London's only less than 5%.

amount of urban space to automobile traffic [2,3]. From a network perspective, this space can be described as the street layer of a larger mathematical object, the multiplex transport network [4–6]. A city's multiplex transport network contains other network layers that have co-evolved with the street layer, such as the bicycle layer or the rail network layer (figure 1). Owing to the car-centric development of most cities, street layers are the most developed layers and define or strongly limit other layers: for example, pavements are by definition footpaths along the side of a street and make up a substantial part of a city's pedestrian space [2]; similarly, most bicycle paths are part of a street or are built along the side.

From an urban sustainability perspective, this situation is suboptimal because the unsustainable mode of automobile transportation dominates sustainable modes like cycling. Consequently, urban planning movements in a number of pioneering cities are increasingly experimenting with drastic policies, such as applying congestion charges (London) [7] and repurposing or removing car parking (Amsterdam, Oslo) [8–10]. These efforts agree in one common goal, together with the literature on cycling safety [11–14] and with cost–benefit analysis [15]. Protected bicycle lanes need to be extended considerably to create complete bicycle networks that provide a safe infrastructure for cycling citizens. Although scattered efforts in this direction have shown preliminary success, a quantitative framework for developing and assessing systematic strategies is missing [16].

Here we analyse the bicycle infrastructure network of 14 world cities, from leading bicycle-culture countries like The Netherlands, to car-centric countries like Great Britain, the USA or Colombia. We first uncover network fragmentation within the bicycle-dedicated infrastructure. Then, to improve a city's vital dedicated bicycle infrastructure [17–20], we develop algorithms for connecting disconnected graphs based on concrete quality metrics from bicycle network planning [21] and apply them to the empirical bicycle networks via network growth simulations. We find that localized investment into targeted missing links can rapidly consolidate fragmented bicycle networks, allowing to significantly increase their connectedness and directness, with potentially crucial implications for sustainable transport policy planning.

## 2. Data acquisition and network construction

We acquired street and bicycle infrastructure networks from multiple cities around the world using OSMnx [22], a Python library to download and construct networks from OpenStreetMap (OSM). OSMnx simplifies the OpenStreetMap's raw data to retain only nodes at the intersections and dead ends of streets, and the spatial geometry of the edges, generating a length-weighted non-planar directed graph [23]. These datasets are of high quality [24,25] in terms of correspondence with municipal open data [26] and completeness: more than 80% of the world is covered by OSM [27]. In particular, OSM's bicycle layer has better coverage than proprietary alternatives like Google Maps [28]. We collect data from a diverse set of cities to capture different development states of bicycle infrastructure networks; from consolidated networks like Amsterdam and Copenhagen, less

developed ones like Manhattan and Mexico City, to rapidly developing cities like Jakarta and Singapore. The various analysed urban areas and their properties are reported in the electronic supplementary material, table SI.1. Code to replicate our results is available as Jupyter Notebooks (https://github.com/nateraluis/bicycle-network-growth) and has been archived within the Zenodo repository: http://doi.org/10.5281/zenodo.4267871; the data can be downloaded from Harvard Dataverse [29].

We characterize each city street and bicycle infrastructure as a primal network, [30] in which nodes are intersections, while links represent bicycle paths, and designated bicycle infrastructure. This recent approach has been useful to demonstrate how cities grow [31,32], how efficient [33] and dense they are, and to capture the tendency of travel routes to gravitate towards city centres [34]. This network is described by an adjacency matrix $A = \{a_{ij}^{[\alpha]}\}$ where $a_{ij} = 1$ if there is a link between nodes $i$ and $j$ and 0 otherwise.

# 3. Defining bicycle network growth strategies and quality metrics

Across all cities considered, we find that almost all network layers are made up of one giant component, except for the bicycle layer which is always fragmented into many disconnected components (see electronic supplementary material, table SI.1). This discovery is remarkable given that the fragmentation occurs also in bicycle-friendly cities like Copenhagen (figure 1), showing that cycling infrastructure can be suboptimal even in the leading cycling cities on the planet. To quantify such an underdevelopment in the sustainable mobility infrastructure of cycling, we focus on the single layer of bicycle networks and on two well-established metrics in bicycle infrastructure quality assessment [21,35–38]: *connectedness* and *directness*. Connectedness indicates 'the ease with which people can travel across the transportation system' [21], and it is related to answering the question 'can I go where I want to, safely?'. Directness addresses the question 'how far out of their way do users have to travel to find a facility they can or want to use?', and can be measured by how easy it is to go from one point to another in a city using bicycle infrastructure versus other mobility options, like car travel.

As our main approach, we choose to measure connectedness and directness over the designated bicycle infrastructure only, without considering travel on streets. Although it is possible to cycle on streets, growing evidence from bicycle infrastructure and safety research is unveiling serious safety issues for cycling when mixed with vehicular traffic [11–13]. However, we also tested our algorithms on a combination of bicycle infrastructure plus streets for which the maximum speed is 30 km h$^{-1}$, following common best-practice reasoning that low speed limits can make streets safe for cycling [39]. The results of these additional simulations are available in the electronic supplementary material; they do not differ significantly from the case of designated bicycle infrastructure presented below, as the developed algorithms follow the same rules to connect the multiple components in both cases of segregated bicycle infrastructure only and of included bikeable streets.

To quantify connectedness, we first measure the number of disconnected components of each city's bicycle network. It is no surprise that car-centric cities have a highly fragmented bicycle infrastructure: for example, London has more than 3000 disconnected bicycle infrastructure segments. However, even bicycle-friendly cities like Copenhagen have over 300 disconnected bicycle path components—see the connected component size distribution $P(N_{cc})$ in figure 1. This infrastructure fragmentation in the bicycle layer poses a challenge for a city's multimodal mobility options [40] and for the safety of its cycling citizens [41,42].

There are various approaches in developing automated strategies for bicycle infrastructure planning. Hyodo *et al.* [43] have proposed a bicycle route choice model to plan bicycle lanes taking into account facility characteristics. Other studies have used input data from bicycle share systems [44] or origin destination matrices [45] to plan bicycle lanes. More recently, taxi trips have been used to identify susceptible clusters for bicycle infrastructure [46]. Here we attempt an alternative approach: since hundreds of bicycle network components already exist in most cities, we aim at consolidating the existing infrastructure by making strategic connections between components rather than starting from scratch.

Our approach takes into account the currently available bicycle infrastructure and uses an algorithmic process to improve the network by finding the most important missing links step by step. This way we focus on optimizing the connectedness metric, growing the bicycle infrastructure by making it more connected, merging parts into fewer and fewer components. We develop two iterative greedy algorithms that we check against a random and a minimum investment approach. The first algorithm, *Largest-to-Second* (L2S), identifies in each step the largest connected component in the bicycle infrastructure network and connects it to the second largest. The second algorithm, *Largest-to-Closest* (L2C), also identifies the largest connected component, but connects it to the closest of the remaining bicycle infrastructure components.

See the electronic supplementary material for details. In both algorithms, components are connected through a direct link between their two closest nodes. We use this technique as an approximation to the underlying street-shortest path—since the most relevant shortest 100 connections typically range from 14 to 500 m, roughly the length of two blocks, this approximation is reasonable. The algorithms repeat this process until there are no more disconnected components in the network.

To have a random baseline, we compare our algorithms with a *Random-to-Closest* (R2C) component approach. In each step of this baseline approach, one component is picked at random and connected with the closest remaining one. This baseline allows us to model a scenario where infrastructure is developed following a systematic but random linking approach—in urban development this corresponds to uncoordinated local planning that randomly connects close pieces of bicycle infrastructure. We also implement a second baseline, the extreme case of *Closest-Components* (CC), which prioritizes connecting the closest two components disregarding their size. This CC approach is equivalent to an 'invest as little as possible' development strategy—it builds up a minimum-spanning-tree-like structure following a modified Kruskal's algorithm [47]. All four algorithms connect components optimizing a well-defined criterion, finding the critical missing links in the network, and adding one new link per iteration. See figure 2a for a schematic of the four algorithms.

We apply the algorithms to the bicycle infrastructure inside the political demarcation of the cities; however, it is possible to extend the methods and include bicycle highways and cross-city trails, since they use as input a set of spatial network components to connect. We opt to not include cross-city links, since they are a special case only available in a few regions and where adequate intra-urban bicycle infrastructure has already been established [48,49].

To test how much cities improve their bicycle layers using these four algorithms, we define two metrics on the bicycle layer that operationalize the notion of connectedness: (i) $n_{LCC} = N_{LCC}/N$, the fraction of nodes from the bicycle infrastructure inside the largest connected component ($N_{LCC}$) compared with the total number of nodes from the same type of infrastructure ($N$), and (ii) $\ell_{LCC} = L_{LCC}/L$, the fraction of link kilometres inside the bicycle infrastructure largest connected component ($L_{LCC}$) compared with the total number of link kilometres in the bicycle network ($L$). Both metrics take values between 0 and 1, where 1 means that there is only one connected component. An intermediate value, for example 0.2, means that the largest connected component contains 20% of all bicycle intersections or path kilometres. Executing our algorithms step by step these metrics can only grow, approaching 1 when the process is complete and they terminate. What distinguishes the algorithms is *how fast* these values grow.

We quantify directness through the metric: (iii) bicycle-car directness $\Delta$, which answers the question 'how direct are the average routes of bicycles compared with cars?' via the ratio between average distance by car and average distance by bicycle. For example, if the shortest car-route from west to east Manhattan is 4 km and the shortest route on the bicycle network between these two points is 5 km, the bicycle-car directness is $4/5 = 0.8$. Note that if the bicycle network is a subset of the street network, then $\Delta$ cannot be larger than 1. Formally, we write $\Delta = \langle \delta_{ij}^b \rangle_{ij} / \langle \delta_{ij}^s \rangle_{ij}$, where $\langle \delta_{ij}^s \rangle_{ij}$ is the average car-route distance, and $\langle \delta_{ij}^b \rangle_{ij}$ is the average length of the shortest bike-route between $i$ and $j$. In each iteration of any of our algorithms, we implement this measure by randomly selecting 1000 pairs of origin–destinations nodes and then averaging the corresponding street/bicycle distance. To avoid undefined values due to disconnected components in the bicycle layer, we add the following condition: if a node from the pair $i$ and $j$ is in a different component, we assign the value $\delta_{ij}^b = 0$. This condition also ensures consistency of growing directness values while the algorithm merges more and more nodes into the same component.

Finally, in order to measure the cumulative efficiency of our algorithms, we define the metric: (iv) $G_{LCC}$ as the relative gain of bicycle path kilometres in the largest connected component. For example, $G_{LCC} = 1.5$ means that the algorithm has increased the largest connected component's original size by 150%. Formally, $G_{LCC} = (L_{LCC} - L_{LCC_0})/L_{LCC_0}$, where $L_{LCC_0}$ is the sum of kilometres in the largest connected component before the algorithm runs. As with all other metrics, $G_{LCC}$ is monotonically increasing with the growth algorithm, and reaches $(1 - \ell_{LCC_0})/\ell_{LCC_0}$ at the end of the dynamics.

# 4. Growing bicycle networks shows stark improvements with small investments

We demonstrate in figure 2 the power of the various growth strategies by showing the initial state of the bicycle layer for the case of Budapest and its state after 85 iterations of the *Largest-to-Closest* algorithm: At this point, the network has almost quadrupled the size of its largest connected component (from 82 to 313 km), with a negligible investment of just less than 5 km (corresponding to 1.4% of the previously

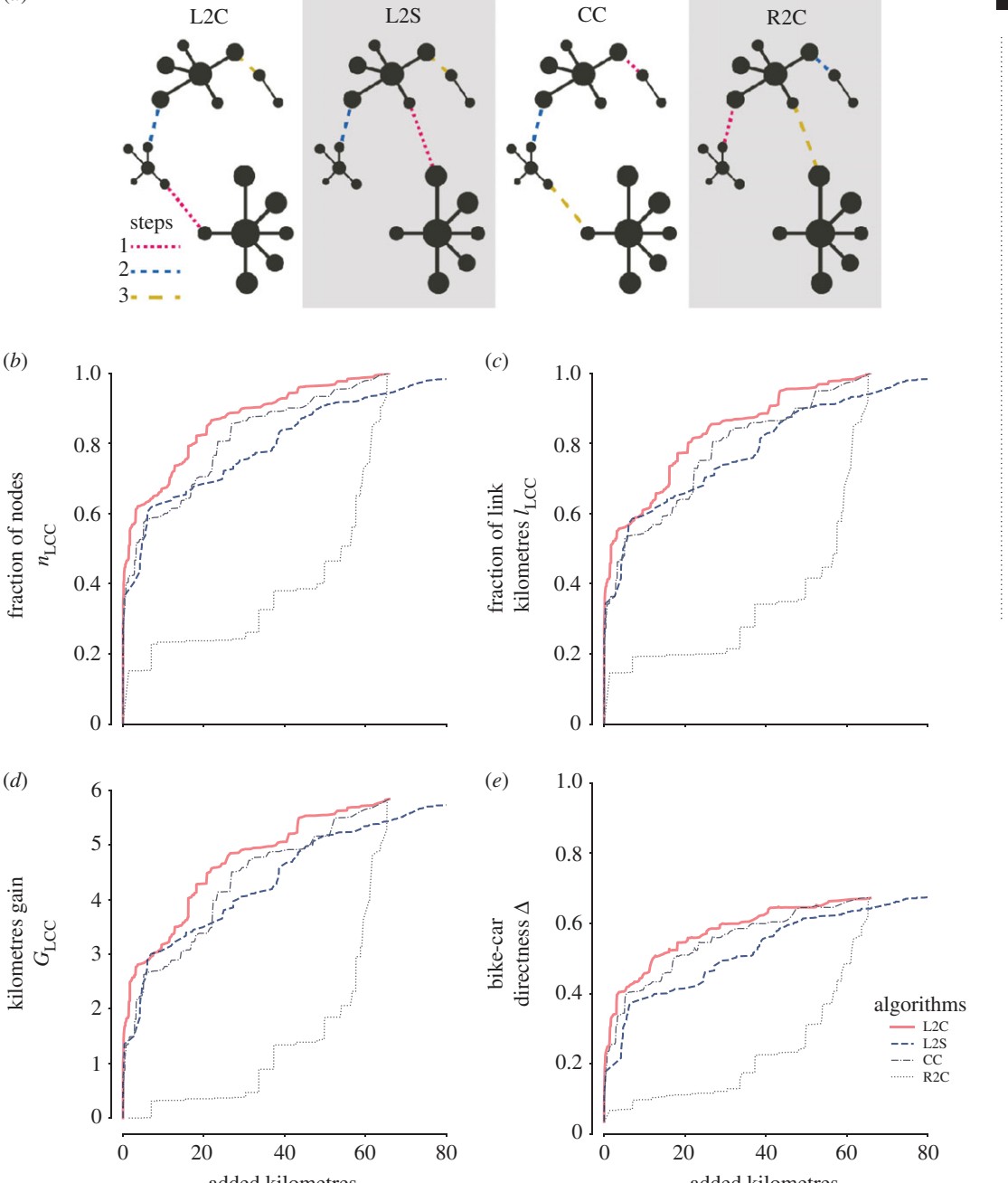

**Figure 2.** (a) Schematic of algorithms to improve bicycle network infrastructure: *Largest-to-Closest* (L2C) finds the largest component and connects it with the closest one; *Largest-to-Second* (L2S) connects the largest component with the second largest; *Closest-Connected* (CC) connects the two closest components; and *Random-to-Closest* (R2C) picks a random component and connects it to the closest. (b) Normalized increase in nodes inside the largest connected component ($n_{LCC}$). (c) Normalized increase in kilometres inside the largest connected component ($\ell_{LCC}$). (d) Kilometres gain ($G_{LCC}$). (e) Bicycle-car directness ($\Delta$). Measures in (b–e) are plotted as a function of the sum of added links in kilometres, for the case of Budapest (for all cities, see electronic supplementary material, figure SI.1).

existing bicycle infrastructure) in new connecting bicycle paths. In terms of connectedness, it goes from 15 to 56% connected. This rapid increase shows that the city can easily improve its bicycle infrastructure with small investments. For some extreme cases, like Bogota, with the same 5 km investment (an increase of 1.3% to the previously existing infrastructure) the bicycle-car directness increases from 6% to almost 48% and connectedness from 34 to 89%. Similar encouraging results hold for other cities, and for all the cities when taking into account the combination of bicycle infrastructure and safely bikeable streets (less than or equal to 30 km h$^{-1}$; see electronic supplementary material).

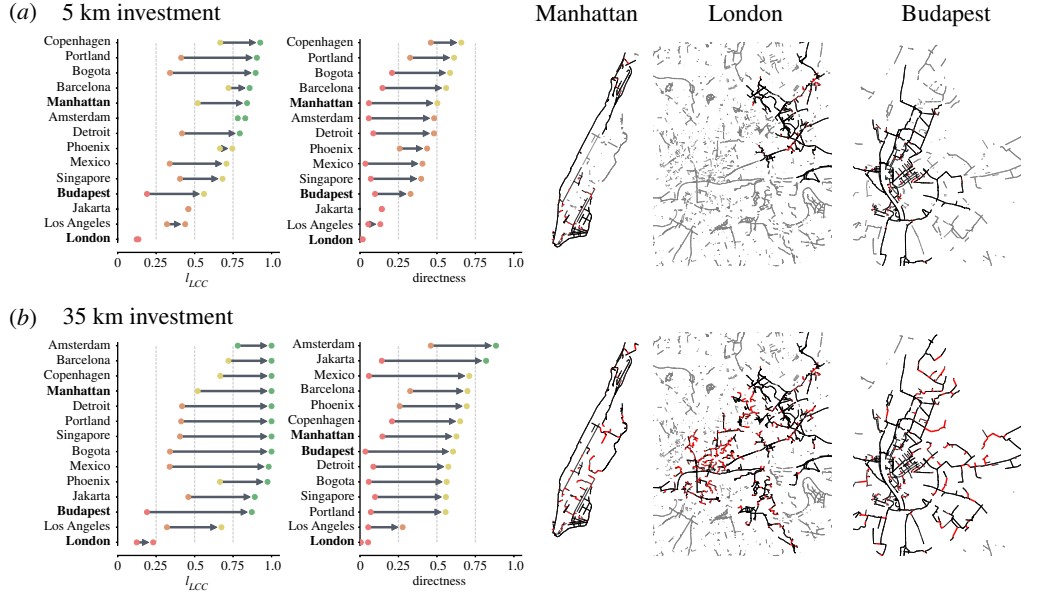

**Figure 3.** Cities improvement and ranking using the *Largest-to-Closest* algorithm. We report the improvement and ranking on the fraction of total kilometres of bicycle infrastructure in the largest connected component ($\ell_{LCC}$) and in the bicycle-car directness ($\Delta$). Dotted lines show thresholds of 25%, 50% and 75%. Plots (*a,b*) show investment strategies of 5 km and 35 km, respectively, the Manhattan, London and Budapest plots show the suggested new links (red) after adding 5 km and 35 km, the newly created largest connected component (black), and the remaining separated components (grey). Videos showing the algorithm evolution are available as electronic supplementary material, Data.

The fraction of nodes inside the largest connected component increases rapidly with newly added links for all considered algorithms except *Random-to-Closest*, figure 2*b*. The *Largest-to-Closest* algorithm performs better than the others, even more than *Closest-Components* which prioritizes minimum investments in the network. Since we are considering bicycle infrastructure, a better practical measure than the number of intersections is the number of kilometres that can be cycled using only designated paths. Figure 2*c* shows how this measure improves in a similarly explosive way: with an investment of only 20 km (5.9% of the existing infrastructure), the largest connected component will contain 80% of the original bicycle infrastructure. Results for the kilometre gain $G_{LCC}$ are shown in figure 2*d*. Three of the four algorithms rapidly gain new kilometres, but as the invested new kilometres grow, each algorithm follows a different gain rate. Also for this metric, *Largest-to-Closest* is the algorithm with the best performance.

We also measure the bicycle-car directness ratio, figure 2*e*. The bicycle-car directness $\Delta$ improves as the algorithms consolidate the network. These improvements are, however, indirectly driven by the improvement of connectedness, which boosts the accessibility of bicycles to different areas of the city. The flattening of the curves at a value considerably smaller than 1 (around 0.65) shows that cars will always outperform bicycles in terms of directness, having on average at least 33% shorter paths in the city. This suboptimal flattening is a natural consequence of the algorithms optimizing for connectedness only, not adding 'redundant' connections. Nevertheless, the measure shows that, similar to connectedness, with a relatively negligible investment of bicycle path kilometres into the system, the bicycle network's directness improves drastically, even in the greediest case where the shortest possible missing link is added in every iteration. This result holds for all analysed cities (see electronic supplementary material). The large differences between the baseline *Random-to-Closest* and our two algorithms (*Largest-to-Second* and *Largest-to-Closest*) show the importance of following an approach that consolidates and grows the largest connected component.

# 5. Different cities have different optimal investment strategies

Differences arise in the state of the bicycle layer and its improvement after applying a growth algorithm. To see this effect, we rank how cities improve using the *Largest-to-Closest* algorithm in two different investment scenarios: investing either (i) 5 km, or (ii) 30 km. Figure 3*a* shows how cities improve when investing 5 km of bicycle infrastructure. We see that some cities get above 75% of their existing

infrastructure connected, meaning that their bicycle layer only needs a small extension. On the other hand, cities like London, Los Angeles and Jakarta need a larger investment to improve. Concerning bicycle-car directness, cities reach lower values due to the focus of the algorithms on completeness. In the worst performing cities like Los Angeles, a covered length close to 50% can be reached easily, while the bicycle-car directness ratio stays below 20%, showing that it is much harder to gain an acceptable bicycle infrastructure in cities where cars are overprioritized. The 35 km investment strategy shows that most cities can get at least 75% of their bicycle infrastructure connected, figure 3*b*. The worst performing outlier is London, due to its bicycle layer containing more than 3000 connected components scattered around 1600 km$^2$ (see electronic supplementary material, table SI.1). In terms of bicycle-car directness London also performs badly, while Amsterdam is the best performing one.

The four proposed metrics capture the impact of newly created connections on the various components of the bicycle network. By linking previously disconnected neighbourhoods with a sustainable mode of transport, our approach focuses on consolidating bicycle infrastructure networks, thus making cities more cohesive and green. It does not, however, focus on growing the bicycle network into large areas of the city not currently served. To test to which extent such connectedness-based algorithms bring an indirect benefit for coverage, we measured the proportion of the city that is covered and reachable by bicycle with an epsilon of 500 m around the bicycle infrastructure of the largest connected component, and calculated the percentage of nodes in the street layer that are covered by the bikeable area. The results of this measurement show a wide range of effects: cities with an already high coverage above 80% (Amsterdam, Copenhagen) reach near instantly 100%, cities with an intermediate coverage (Manhattan, Bogota, Budapest) follow a more linear progression per added kilometre, while underdeveloped or sprawling cities (LA, London, Jakarta) show negligible growth (electronic supplementary material, figure SI.7). While our present goal (consolidation of the bike network) is intended to show the potential of our approach, an extension towards the exploration of new city areas will increase further the real-world applicability of our results. We consider this extension an interesting line of future research in the challenge of developing optimal data-driven strategies of transport network growth, potentially informed by theoretical frameworks such as optimal percolation [50,51]. Besides, the use of other network metrics, such as network efficiency [52], might unveil new dimensions characterizing the impact of the proposed algorithms on the development of the bicycle infrastructure.

# 6. Discussion

Our starting point showed that a common characteristic of cities is the fragmentation of their bicycle networks. We have proposed the use of data-driven algorithms to consolidate bicycle network components into connected networks to improve efficiently sustainable transport. We have shown that connecting the bicycle infrastructure in an algorithmic way rapidly improves the connectedness and directness of the bicycle layer. These algorithms, when compared with two baselines, highlight the usefulness of growing the bicycle network on a city-wide scale (considering all areas of the city) rather than randomly adding local bicycle infrastructure. Improving the connectivity of bicycle lanes and paths improves not only the network itself, but also promotes the use of bicycles as means of transportation in a city, improving the health of its inhabitants [53].

Improving bicycle infrastructure one link at the time (by identifying suitable components to connect) is only the first step towards a systematic framework for realistic bicycle network growth strategies. Our current approach is not the last word in this development, since it does not yet explicitly optimize for directness and does not account for transport flow. Further, our proposed approach helps implement a more connected transport network which can improve the possibilities for multimodal transport. This could be the starting point for implementing truly multimodal strategies, such as integration with public transportation, or bicycle parkings in transportation hubs [21].

In our algorithms, each new link works as a bridge between components, potentially having large betweenness centrality. Such high-betweenness segments could become overused and create bottlenecks in practice. To improve this situation, it would be necessary to create links in the network that act as redundant paths. In doing so, directness and coverage would also be improved, along with the network's robustness to interruptions. This is an interesting and possibly demanding task that we leave for future research, as the new links would have to be created in a coherent manner balancing trade-offs between network structure and mobility dynamics. We anticipate that complementing OpenStreetMap data with additional information on the use of traffic flow and movement data, like

trips from bike share systems or origin–destination matrices, possibly from alternative sources such as municipalities and transportation agencies, might further improve the algorithms by better detecting underserved and optimal areas in the city where new links should be created. Despite these various possibilities for qualitative updates to the studied growth strategies, our first models have demonstrated the capability to generate substantial improvements with minimal effort.

The use of data-driven algorithms to identify crucially missing links in bicycle infrastructure has the potential to improve the mobility infrastructure of cities efficiently and economically. This approach is not only useful for planning city structure, but could also be used together with simulating mobility flows and to provide insights on how the system will behave after new measures are implemented. Ultimately, planning cultures and processes will also have to be accounted for [54]. We anticipate that a future stream of work should include longitudinal studies [55] in multiple cities, along with algorithmic simulations to first model and simulate possible changes to the transport network, and then to test those models with ground truth data, to compare the evolution of infrastructure and mobility dynamics between cities with different transport priorities.

Data accessibility. Relevant code for this research work is stored in GitHub: https://github.com/nateraluis/bicycle-network-growth and have been archived within the Zenodo repository: http://doi.org/10.5281/zenodo.4267871. The data can be downloaded from Harvard Dataverse (https://doi.org/10.7910/DVN/GSOPCK).

Authors' contributions. L.G.N.O., F.B. and M.S. conceived the research. L.G.N.O. collected, processed and cleaned data, and carried out the computational analysis. L.G.N.O., F.B., G.I and M.S. designed the algorithms, analysed the data and results, and helped draft the manuscript. L.G.N.O., F.B., G.I and M.S. gave final approval for publication and agree to be held accountable for the work performed therein.

Competing interests. We declare we have no competing interests.

Funding. G.I. acknowledges support by the European Commission through H2020 projects HumanE AI (G.A. no. 761758) and HumanE AI Net (G.A. no. 952026), and from the Air Force Office of Scientific under award no. FA8655-20-1-7020.

Acknowledgements. The authors wish to thank Ana Paula Velasco and Roberta Sinatra for their comments and support. We are also grateful to the anonymous reviewers for their helpful comments.

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
