## [Reviewer comments · Royal Society Open Science]

Review History

RSOS-201130.R0 (Original submission)

Review form: Reviewer 1 (Samuel Johnson)

Is the manuscript scientifically sound in its present form?

Yes

Are the interpretations and conclusions justified by the results?

Yes

Is the language acceptable?

Yes

Do you have any ethical concerns with this paper?

No

Have you any concerns about statistical analyses in this paper?

No

Recommendation?

Accept with minor revision (please list in comments)

Comments to the Author(s)

Please see attached document (Appendix A).

Review form: Reviewer 2

Is the manuscript scientifically sound in its present form?

Yes

Are the interpretations and conclusions justified by the results?

Yes

Is the language acceptable?

Yes

Do you have any ethical concerns with this paper?

No

Have you any concerns about statistical analyses in this paper?

No

Recommendation?

Accept with minor revision (please list in comments)

Comments to the Author(s)

The authors study the OpenStreetMap cycle lane networks of several cities around the world, where nodes are intersections and links are cycle lane segments. First, they reveal that most networks consist of a number of disconnected components. Secondly, they propose two algorithms for building new cycle lane segments in a principled way, in order to improve the quality of the network with minimal cost. They quantify the evolution of the network quality (in terms of connectedness and in comparison to the car network) as a function of segments added for different algorithm. Ultimately, they identify a strategy for building new cycle lane segments that is much more effective than a random strategy.

The topic presented in the paper is very relevant for policy makers and a good fit for the interdisciplinary readership of RSOS. The article is clear in scope and the analyses are rigorous. I can see the paper has substantially improved following the previous rounds of revisions. I recommend publication pending the following minor comment is addressed.

Judging from Figure R3, it seems plausible that some of the added links could be missing in the networks due to limited data quality. I can see that the authors discuss data quality in the data description, and mention that OpenStreetMap data is estimated to cover the 80% of the network. For the type of analysis presented in this paper, the missing 20% can make some difference. If the authors find the data, I think the paper would benefit from a comparison of the results obtained using OpenStreetMap data and official data obtained from some municipalities. If not possible, the authors should discuss in the conclusions this potential limitation and tone down sentences like . "This discovery is remarkable given that the fragmentation occurs also in bicycle-friendly cities like Copenhagen (Fig. 1), showing that cycling infrastructure is suboptimal even in the best cycling cities on the planet."

Decision letter (RSOS-201130.R0)

Dear Mr Natera Orozco

On behalf of the Editors, we are pleased to inform you that your Manuscript RSOS-201130 "Data-driven strategies for optimal bicycle network growth" has been accepted for publication in Royal Society Open Science subject to minor revision in accordance with the referees' reports. Please find the referees' comments along with any feedback from the Editors below my signature.

Please submit your revised manuscript and required files (see below) no later than 7 days from today's (ie 13-Oct-2020) date. Note: the ScholarOne system will 'lock' if submission of the revision is attempted 7 or more days after the deadline. If you do not think you will be able to meet this deadline please contact the editorial office immediately.

Kind regards,

Anita Kristiansen
Editorial Coordinator

on behalf of Dr Robert MacKay (Associate Editor) and Mark Chaplain (Subject Editor)
openscience@royalsociety.org

Associate Editor Comments to Author (Dr Robert MacKay):

Comments to the Author:

The reviewers are both positive but suggest some further revisions. I agree that the issues they raise need addressing before we could publish. Furthermore, one reviewer noted (in a separate report for the editors) that the data were not accessible. This needs rectifying.

Reviewer comments to Author:

Reviewer: 1

Comments to the Author(s)

Please see attached document.

Reviewer: 2

Comments to the Author(s)

The authors study the OpenStreetMap cycle lane networks of several cities around the world, where nodes are intersections and links are cycle lane segments. First, they reveal that most

networks consist of a number of disconnected components. Secondly, they propose two algorithms for building new cycle lane segments in a principled way, in order to improve the quality of the network with minimal cost. They quantify the evolution of the network quality (in terms of connectedness and in comparison to the car network) as a function of segments added for different algorithm. Ultimately, they identify a strategy for building new cycle lane segments that is much more effective than a random strategy.

The topic presented in the paper is very relevant for policy makers and a good fit for the interdisciplinary readership of RSOS. The article is clear in scope and the analyses are rigorous. I can see the paper has substantially improved following the previous rounds of revisions. I recommend publication pending the following minor comment is addressed.

Judging from Figure R3, it seems plausible that some of the added links could be missing in the networks due to limited data quality. I can see that the authors discuss data quality in the data description, and mention that OpenStreetMap data is estimated to cover the 80% of the network. For the type of analysis presented in this paper, the missing 20% can make some difference. If the authors find the data, I think the paper would benefit from a comparison of the results obtained using OpenStreetMap data and official data obtained from some municipalities. If not possible, the authors should discuss in the conclusions this potential limitation and tone down sentences like . "This discovery is remarkable given that the fragmentation occurs also in bicycle-friendly cities like Copenhagen (Fig. 1), showing that cycling infrastructure is suboptimal even in the best cycling cities on the planet."

===PREPARING YOUR MANUSCRIPT===

- one version identifying all the changes that have been made (for instance, in coloured highlight, in bold text, or tracked changes);
- a 'clean' version of the new manuscript that incorporates the changes made, but does not highlight them. This version will be used for typesetting.

===PREPARING YOUR REVISION IN SCHOLARONE===

To revise your manuscript, log into <https://mc.manuscriptcentral.com/rsos> and enter your Author Centre - this may be accessed by clicking on "Author" in the dark toolbar at the top of the

page (just below the journal name). You will find your manuscript listed under "Manuscripts with Decisions". Under "Actions", click on "Create a Revision".

<https://royalsociety.org/journals/authors/author-guidelines/#supplementary-material> to include a suitable title and informative caption. An example of appropriate titling and captioning may be found at https://figshare.com/articles/Table_S2_from_Is_there_a_trade-off_between_peak_performance_and_performance_breadth_across_temperatures_for_aerobic_sc_ope_in_teleost_fishes_/3843624.

Author's Response to Decision Letter for (RSOS-201130.R0)

See Appendix B.

Decision letter (RSOS-201130.R1)

Dear Mr Natera Orozco,

It is a pleasure to accept your manuscript entitled "Data-driven strategies for optimal bicycle network growth" in its current form for publication in Royal Society Open Science. The comments of the reviewer(s) who reviewed your manuscript are included at the foot of this letter.

on behalf of Dr Robert MacKay (Associate Editor) and Mark Chaplain (Subject Editor)
openscience@royalsociety.org

Appendix A

Data-driven strategies for optimal bicycle network growth

The authors build a network of cycle lanes from OpenStreetMap data for each of several cities. They then propose and apply greedy algorithms for introducing new edges into the networks, and analyse the effect on two measures of “connectedness” (the fractions of nodes and of edges in the giant connected component (GCC)), and one of “directness” (a comparison of shortest distance by car and by bike). They show that algorithms which make use of the size of connected components to be linked outperform those which only use distance between components; and that in some cities a relatively small investment in new bike lanes can increase the connectedness of the network significantly.

The manuscript is well written and the results clearly presented. Given the benefits to society which might ensue from improving cycling infrastructure in cities, I believe this work presents a useful new approach to a pressing real-world problem. I have just three concerns about the work. I appreciate, however, that the authors have already made significant changes to their manuscript in response to three previous reviews. I therefore suggest how these concerns might be met to some extent with relatively minor changes rather than redoing much of the work.

My main concern is that barely no account appears to be taken of the relative size of different cities. It seems to me that this leads to an unnecessarily pessimistic outlook for large cities like London or Los Angeles. For example, Fig. 3 shows the “Cities [sic] improvement and raking using the Largest-to-Closest algorithm”, after either an extra 5 km or 35 km of cycle lanes are added. The figure and accompanying text highlight the cases of Manhattan, Budapest and London, showing that the former two can improve significantly with such investment, while London cannot. But a Google search tells me that the surface areas of these “cities” are 59, 525 and 1,572 km², respectively. Given such large disparities in size, it would seem more appropriate to use a relative measure of extra cycle lane investment, rather than the absolute number of kilometres. Unfortunately, “added kilometres” is the independent variable used for almost all the results in the paper.

I suggest that the authors at least include a table with the total length of cycle lanes for each city (Table SI.1 lists some information about each network, but I don’t believe this is enough to make sense of the addition of new edges in terms of extra kilometres). For comparison, the total length of roads for cars might also be useful. I also believe the work would benefit from the addition of figures like Fig. SI. 2-10, but with the x-axis rescaled by total cycle-lane length – i.e. instead of “Added kilometres”, “Added fraction of extant kilometres”. I assume this is relatively easy to do without needing to rerun the algorithms, since it is just a question of plotting the same outputs against a rescaled variable. However, perhaps in the cases of large cities the maximum kilometres added would need to be higher for a good comparison.

A less important but related point is that defining connectedness in terms only of the GCC may not be the most informative approach for large cities. For instance, while one might happily cycle across Amsterdam or Copenhagen to get to work every day, few Londoners are likely to cycle the 35 km from Walthamstow to Wembley. So perhaps a measure such as network efficiency (i.e. the mean of the inverse shortest paths) would better capture the ease of travel between parts of the city, especially if one wishes to compare cities with sizes differing in orders of magnitude.

I don’t propose, however, that the authors redo their work using a new measure of connectedness; I believe it would suffice to mention that the same algorithms could be applied using different metrics if desired, such as efficiency.

Finally, there is no discussion of the process of cycle-lane consolidation in terms of percolation. However, there is presumably a transition from a situation in which the network is fragmented in

many small components, to one in which most nodes and edges belong to the GCC. In fact, from the vaguely power-law distribution of sizes seen in Fig. SI.1 for some cities, it may be that these are close to the percolation transition. If this is the case, perhaps one could use the authors' methods to quantify the minimum investment needed to achieve percolation. On the other hand, the proposed algorithms are a kind of inverse Achiloptas process: edges are placed preferentially between large components. This, and the spatial structure of the networks, may mean that the transition is less pronounced and hence of less importance than might be expected. Either way, I believe the paper would benefit from some comment on this point.

Appendix B

Dear Reviewers,

Please find enclosed the revised version of our paper “Data-driven strategies for optimal bicycle network growth” by Natera et al. We have carried out a revision accounting for the remarks of the two referees and editor.

We provide in the following a detailed answer to the criticisms raised by both referees. We are very satisfied with this revised version, and we hope that this revised manuscript is in its final shape ready for publication in the Journal of the Royal Society Open Science.

Reviewer #1

We would like to thank the reviewer for a very positive assessment of our work. We have considered the reviewer’s comments carefully and included them in the revised version of our manuscript.

R1: (1) My main concern is that barely no account appears to be taken of the relative size of different cities. It seems to me that this leads to an unnecessarily pessimistic outlook for large cities like London or Los Angeles. For example, Fig. 3 shows the “Cities [sic] improvement and raking using the Largest-to-Closest algorithm”, after either an extra 5 km or 35 km of cycle lanes are added. The figure and accompanying text highlight the cases of Manhattan, Budapest and London, showing that the former two can improve significantly with such investment, while London cannot. But a Google search tells me that the surface areas of these “cities” are 59, 525 and 1,572 km, respectively. Given such large disparities in size, it would seem more appropriate to use a relative measure of extra cycle lane investment, rather than the absolute number of kilometres. Unfortunately, “added kilometres” is the independent variable used for almost all the results in the paper.

(1.1) ► We thank the reviewer for raising the point regarding the cities’ sizes. Indeed, they span across various urban scales, enabling us to test our methods in multiple scenarios, from sprawling cities like London or Los Angeles, to more compact cities like Budapest or Amsterdam. In the same sense the variable “added kilometers” appeals to urban planners and city decision makers as an indicator to understand the size of the necessary investment in new infrastructure to improve the conditions of the city.

Following the reviewer’s suggestion we have modified parts of the text to reflect the percentage that those 5 or 35 new kilometers represent to the cities.

At this point the network has almost quadrupled the size of its largest connected component (from 82 km to 313 km), with a negligible investment of just less than 5 km (corresponding to 1.4% of the previously existing bicycle infrastructure) in new connecting bicycle paths. In terms of connectedness, it goes from 15% to 56% connected. This rapid increase shows that the city can easily improve its bicycle infrastructure with small investments. For some extreme cases, like Bogota, with the same 5 km investment (an increase of 1.3% to the previously existing infrastructure) the bicycle-car directness increases from 6% to almost 48% and connectedness from 34% to 89%.

[...] with an investment of only 20 km (5.9% of the existing infrastructure), the largest connected component will contain 80% of the original bicycle infrastructure.

R1: (2) I suggest that the authors at least include a table with the total length of cycle lanes for each city (Table SI.1 lists some information about each network, but I don’t believe this is enough to make sense of the addition of new edges in terms of extra kilometres). For comparison, the total length of roads for cars might also be useful. I also believe the work would benefit from the addition of figures like Fig. SI. 2-10, but with the x-axis rescaled by total cycle-lane length – i.e. instead of “Added kilometres”, “Added fraction of extant kilometres”. I assume this is relatively easy to do without needing to rerun the algorithms, since it is just a question of plotting the same outputs against a rescaled

variable. However, perhaps in the cases of large cities the maximum kilometres added would need to be higher for a good comparison.

(1.2) ► We thank the reviewer for the suggestion to include the total length of cycle lanes in Table SI. 1. We have not only included the length of bicycle infrastructure, but also added the lengths for the other infrastructures as a reference to understand the state of city development and compare sizes between cities.

Regarding the figures, we now have included a new figure to the Supplementary Information in which we plot the increase of kilometers against the “Added fraction of extant kilometers”.

Figure R1: Normalized increase in kilometers inside the largest connected component (ℓ_{LCC}) versus the fraction of extant kilometers to be added.

R1: (3) A less important but related point is that defining connectedness in terms only of the GCC may not be the most informative approach for large cities. For instance, while one might happily cycle across Amsterdam or Copenhagen to get to work every day, few Londoners are likely to cycle the 35 km from Walthamstow to Wembley. So perhaps a measure such as network efficiency (i.e. the mean of the inverse shortest paths) would better capture the ease of travel between parts of the city, especially if one wishes to compare cities with sizes differing in orders of magnitude.

I don't propose, however, that the authors redo their work using a new measure of con-

nectedness; I believe it would suffice to mention that the same algorithms could be applied using different metrics if desired, such as efficiency.

(1.3) ► We thank the reviewer for the suggestion and agree with such assessment. The algorithms can be evaluated using different measures, in this case the network efficiency can be one of those. We have modified the following text to reflect this comment:

Besides, the use of other network metrics, such as network efficiency [52], might unveil new dimensions characterising the impact of the proposed algorithms on the development of the bicycle infrastructure.

The corresponding reference is:

[52] Vito Latora and Massimo Marchiori. Efficient behavior of small-world networks. *Phys. Rev. Lett.*, 87:198701, Oct 2001.

R1: (4) **There is no discussion of the process of cycle-lane consolidation in terms of percolation. However, there is presumably a transition from a situation in which the network is fragmented in many small components, to one in which most nodes and edges belong to the GCC. In fact, from the vaguely power-law distribution of sizes seen in Fig. SI.1 for some cities, it may be that these are close to the percolation transition. If this is the case, perhaps one could use the authors' methods to quantify the minimum investment needed to achieve percolation. On the other hand, the proposed algorithms are a kind of inverse Achiloptas process: edges are placed preferentially between large components. This, and the spatial structure of the networks, may mean that the transition is less pronounced and hence of less importance than might be expected. Either way, I believe the paper would benefit from some comment on this point.**

(1.4) ► We agree with the reviewer. We have now included the following text in the revised manuscript:

While our present goal (consolidation of the bike network) is intended to show the potential of our approach, an extension towards the exploration of new city areas will increase further the real-world applicability of our results. We consider this extension an interesting line of future research in the challenge of developing optimal data-driven strategies of transport network growth, potentially informed by theoretical frameworks such as optimal percolation [50, 51].

Corresponding references are:

[50] Dimitris Achlioptas, Raissa M. D'Souza, and Joel Spencer. Explosive percolation in random networks. *Science*, 323(5920):1453–1455, 2009.

[51] Flaviano Morone and Hernán A. Makse. Influence maximization in complex networks through optimal percolation. *Nature*, 524(7563):65–68, 2015.

Reviewer #2 We would like to thank the reviewer for a thorough and critical reading of our manuscript. We have done our best to include the reviewer's comments in its revised version.

R2: Judging from Figure R3, it seems plausible that some of the added links could be missing in the networks due to limited data quality. I can see that the authors discuss data quality in the data description, and mention that OpenStreetMap data is estimated to cover the 80% of the network. For the type of analysis presented in this paper, the missing 20% can make some difference. If the authors find the data, I think the paper would benefit from a comparison of the results obtained using OpenStreetMap data and official data obtained from some municipalities. If not possible, the authors should discuss in the conclusions this potential limitation and tone down sentences like. “This discovery is remarkable given that the fragmentation occurs also in bicycle-friendly cities like Copenhagen (Fig. 1), showing that cycling infrastructure is suboptimal even in the best cycling cities on the planet.”

(2.1) ► We thank the reviewer for raising the point about the possible impact of the missing data when working with OpenStreetMap. We agree with the assessment: the impact of the possible missing data can make some difference, especially for the bicycle infrastructure. Unfortunately, we were not able to find municipal data to provide a comparison between data sources.

We have modified the sentence:

This discovery is remarkable given that the fragmentation occurs also in bicycle-friendly cities like Copenhagen (Fig. 1), showing that cycling infrastructure can be suboptimal even in the leading cycling cities on the planet.

We also added the following sentence to the Discussion section:

We anticipate that complementing OpenStreetMap data with additional information on the use of traffic flow and movement data, like trips from bike share systems or origin-destination matrices, possibly from alternative sources such as municipalities and transportation agencies, might further improve the algorithms by better detecting underserved and optimal areas in the city where new links should be created. Despite these various possibilities for qualitative updates to the studied growth strategies, our first models have demonstrated the capability to generate substantial improvements with minimal effort.